# Amphibian Skin and Skin Secretion: An Exotic Source of Bioactive Peptides and Its Application

**DOI:** 10.3390/foods12061282

**Published:** 2023-03-17

**Authors:** Sylvia Indriani, Supatra Karnjanapratum, Nilesh Prakash Nirmal, Sitthipong Nalinanon

**Affiliations:** 1School of Food Industry, King Mongkut’s Institute of Technology Ladkrabang, Ladkrabang, Bangkok 10520, Thailand; 2Professional Culinary Arts Program, School of Management, Walailak University, Nakhon Si Thammarat 80161, Thailand; 3Food Technology and Innovation Research Center of Excellence, Department of Agro-Industry, School of Agricultural Technology, Walailak University, Thasala, Nakhon Si Thammarat 80161, Thailand; 4Institute of Nutrition, Mahidol University, Salaya, Nakhon Pathom 73170, Thailand

**Keywords:** amphibian, frogs, salamanders, skin, skin secretions, bioactive peptides, food consumption, food safety

## Abstract

Amphibians have been consumed as an alternative protein source all around the world due to their delicacy. The skin of edible amphibians, particularly frogs and giant salamanders, always goes to waste without further utilization. However, these wastes can be utilized to extract protein and bioactive peptides (BPs). Various BPs have been extracted and reported for numerous biological activities such as antioxidant, antimicrobial, anticancer, antidiabetic, etc. The main BPs identified were brevinins, bombesins, dermaseptins, esculentins, magainin, temporins, tigerinins, and salamandrins. This review provides a comprehensive discussion on various BPs isolated and identified from different amphibian skins or skin secretion and their biological activities. The general nutritional composition and production statues of amphibians were described. Additionally, multiple constraints against the utilization of amphibian skin and secretions are reported. Finally, the prospective applications of BPs in food and biomedical industries are presented such as multifunctional food additives and/or supplements as well as drug delivery agents.

## 1. Introduction

Amphibians have been studied for many purposes such as for consumption, art, pharmacology, calligraphy, culture, poetry, entertainment, religion, and clinical studies [1,2]. They have been considered and used as a compatible model in discovering new information about all animals [3]. Amphibians have been raised in captivity for human consumption as high-prestige rare meats, particularly frogs (*Rana* sp.) and giant salamanders (*Andrias* sp.) [4,5]. These species have been utilized not only as a delicacy, but also as drugs in Southeast Asia, mostly in China [6,7]. *R. temporaria*, known as Common Frog in Europe, is widely distributed in both open and forested habitats [8]. In general, processing waste or by-products occurs during production, processing, distribution, and marketing to various points of consumption. Skin is one of the typical by-products that can be used as a collagenous source. Frogs and giant salamander wastes (skin) are recognized as alternative collagenous or bioactive peptide (BP) sources, providing good physicochemical characteristics and bioactivities such as antioxidative [9,10], antimicrobial [5,11,12], anti-inflammatory [13], anticancer [12], antidiabetic [14], and drug delivery activities [15]. Several BPs have been isolated and studied from the skin of amphibians such as bradykinin [16], bombesin [17], bombinins [18], caeruleins [19], chromogranins/secretogranins [20], dermaseptins [21], tachykinins [22], and temporins [23]. Tigerinins from *R. tigerina* have been studied for their antimicrobial, wound healing, and antidiabetic properties [14,24]. Recently, Karnjanapratum and Benjakul [25] reported that the skin of an Asian bullfrog was a good collagenous source for preparing a collagenous derivative, which showed a high potential for biocompatible material for human consumption. Amphibians are thermo- and hygric-sensitive species, which enable them to respond to different environments promptly. This physiological ability to adapt can affect the histology of skin matrices and is thus distinct from other animal skins [8]. For instance, frogs and salamanders have a physiological self-defense system when they are in harsh or dangerous conditions where their skin will secrete some bioactive compounds (i.e., antimicrobial and antioxidative compounds) or even toxins [26,27]. 

Although amphibians have been utilized and explored widely, several issues regarding the consumption of amphibians and their derivative products cannot be disregarded including ethical, health, and religious issues [28]. For instance, the exploitation of amphibians should follow the instructions from The International Union for Conservation Nature (IUCN) Red List of Threatened Species. Frog meat is also known as non-halal (haram) for Muslims due to their Islamic dietary laws, called hadith. These typical multiple constraints should be addressed before human consumption and animal welfare to avoid vagueness and overexploitation in their utilization. Therefore, this review discusses the production status and nutritional composition of amphibians. A comprehensive discussion has been conducted on amphibian skin and skin secretion utilization for bioactive peptides. Various identified BPs have been reported for their biological activities. The multiple constraints of amphibian utilization and consumption are also discussed as well as the prospective food and biomedical applications outlined.

## 2. Amphibians Production and Nutritional Composition 

### 2.1. Production Status

Amphibian populations have been declining globally as a result of overexploitation paired with about more than 50% of amphibians listed as vulnerable, endangered, or critically endangered by the IUCN Red List [29]. Although most known edible amphibians are frogs and salamanders, there is limited information regarding their global trade value. An integrated study using market surveys and interviews was conducted to study and discuss the domestic trade of frogs and salamanders. Most previous reports have been focused on frogs rather than salamanders. The overexploitation of salamanders (Chinese giant salamanders) for food has been reported, whereas their natural habitats have been found to be destroyed in the recent decade, particularly in China [30], and was categorized as a Critically Endangered species by the IUCN Red List of Threatened Species. Therefore, conservation has been initiated by collaboration between local government and farmers to fulfill the market demand and reduce overexploitation. For instance, the collaboration took place in Qinling region (China), where the government subsidized the initiation of the farm with an open investment from other locals [30]. On the other hand, frog farming and its culture have been part of Asian countries due to its low investment, relative ease of management, and its nutrition. The biotechnology trend has brought frog meat to popularity in the market, as proven by its increasing production mass [31]. For instance, in Thailand, slaughtered Asian bullfrogs have been exported to Hong Kong, Singapore, Malaysia, and Western countries with a capacity of at least 50 megatons per month. This amount has increased annually due to its high demand [32]. In southwest Nigeria, thirty-two (32) frog collectors were able to harvest a total of 2,738,610 frog species [33]. Frog meat is commonly exported as frozen deskinned legs where the other body parts are discarded without further utilization including the head, trunk, and skin [34,35]. Figure 1 illustrates the general production scheme of edible amphibians for both local consumption (a) and for domestic and global exportation (b). The amphibian by-products consist of skins and other undesirable body parts. For local consumption (Figure 1a), less by-products can be obtained as less processing steps are involved. This is in contrast with those for domestic and global exportation (Figure 1b).

Onadeko, Egonmwan, and Saliu [36] identified the predominant edible frog species in Nigeria as *Hildebrandtia ornate*, *Hoplobatracus occipitalis*, *Ptychadena pumilio*, and *Pyxicephalus edulis.* For non-consumptive utilization (such as medicinal and mythic purposes), *Bufo pentoni*, *Ptychadena mascareniensis*, *Tomoptera cryptotis*, and *Amietophyrynus* regularis have been used. The high annual demand in Nigeria has kept up with the large supply quantities where the excessive supply can be exported. In Indonesia, especially on Java Island, Crab-eating frog (*Fejervarya cancrivora*), Giant javan frog (*Limnonected macrodon*), and Grass frog (*F. limnocharis*), have been reported to contribute to 75.4%, 18.8%, and 5.8% of the total catch, respectively. Based on the Foreign Trade Statistical Bulletin by the Indonesian Statistical Bureau, a fluctuating trend of the export of frogs’ legs was observed from 1969 to 2002 [34]. Gratwicke, Evans, Jenkins, Kusrini, Moore, Sevin, and Wildt [35] reported that Indonesia contributed around half of the global frog leg trade, accounting for USD 40 million where France, Belgium, and the USA were the predominant importers (75%). Unfortunately, no further update has been found since. Belgium, Luxembourg, France, the Netherlands, Singapore, and Hong Kong are the main export destinations. A production obstacle was also reported as during the dry season (February to August), it was sometimes hard to comply with the high demand of frogs’ legs. Kusrini and Alford [34] projected a decline in the annual yield as it was affected by an increasing number of harvesters and middlemen as well as habitat and climate change, as more rice fields have been developed. Thus, it could lead to enlarge the gap between supply and demand. A decline in yield, along with a low export rate, might be explained by a higher rate of amphibian consumption among the locals, since they were familiar with the meat [4]. Therefore, by applying multiple coordinated approaches, their sustainability and productivity can be maintained without abandoning animal welfare, thus reducing the level of overexploitation [1,4,6].

### 2.2. Nutritional Composition

The most common edible amphibians are frogs and salamanders. Frogs’ legs have been known as a unique culinary delicacy in Europe, the USA, Asia, Australia, and Africa [34,36]. It has been consumed as an alternative meat replacing chicken, since its taste is perceived to be similar to chicken. Most developed countries fulfill their frog meat demand by importing it from developing countries including Indonesia, Thailand, and Nigeria [29]. Meanwhile, salamanders are more popular in China than in other countries, particularly Chinese giant salamanders (CGS). The meat of CGS provide a tender texture with a similar taste to fish [7]. Both frogs and salamanders have been consumed due to their comparable nutritional value and eating quality compared to that of other land animals. Table 1 presents the nutritional composition of edible amphibians from previous reports. The diversity of their nutritional values is influenced by different biodiversity, age, sex, feeding, farming, breeding type, and body part [37]. Fresh frog meats (fore-chest, thigh, and calf) consist of various moisture (68.9–83.4 g/100g), protein (10.5–21.2 g/100 g), fat (0.6–5.1 g/100 g), carbohydrate (0.5–4.6 g/100 g), and ash (0.6–5.0 g/100 g). CGS meat consists of moisture (79.0–82.3 g/100 g), protein (14.0–16.4 g/100 g), fat (2.5–3.5 g/100 g), and ash (0.7–1.1 g/100 g), however, their carbohydrate content has not been reported yet. The meat from frog species is comparable with that of CGS. He, Zhu, Zeng, Lin, Ji, Wang, Zhang, Lu, Zhao, Su, and Xing [7] reported that although different cultural environments and varieties of amphibians could yield different amounts, the main composition, nutritional efficacy, and medicinal characteristics were roughly the same. Here, the protein content of CGS was found to be slightly lower than frog meat. Nevertheless, both frogs and CGS meats were comparable with that of chicken and fish per edible portion. Some antinutritive compounds were found in *Pelophylax esculentus* and reported as dry basis matter including saponin (1.8 mg/100 g), tannin (5.4 mg/100 g), flavonoid (1.8 mg/100 g), and oxalate (2.8 mg/100 g) [38]. Thus, this could be a limiting factor in the consumption of edible amphibians.

## 3. Bioactivities of Peptides Derived from Amphibian Skin

Generally, amphibian processing involves deskinning to obtain meat for consumption. As previously reported, high demand will automatically generate a high by-product such as skin. Amphibian skin can offer a good source of protein, basically collagen, which can be converted into BPs and collagenous derivates. BPs from amphibian skins and secretions have at least one bioactivity including antioxidant [27,46,47], antimicrobial [5,26,48], anticancer [49,50], antidiabetic activity [26,51], and others. When amphibians are alive, especially frogs, their skin secretions can be collected and purified to obtain peptides with multiple bioactivities. Previous studies on frog skin secretions have reported the amino acid sequences of related BPs [12,14]. Although a direct association between the amino acid composition and bioactivity of the peptide remains unclear, cationic and hydrophobic peptides are presumed to form an amphipathic helix in a membrane-mimetic environment [50]. Furthermore, collagen and its derivatives from amphibian skins provide similar health benefits to the human body [2,13,15,48]. Amphibian skins have been considered as collagenous sources for the production of collagen, collagen hydrolysate (CH), and gelatin [25,52,53,54]. The biofunctional properties of peptides involve serial complex biochemistry reactions and mechanisms. These lead to various actions performed by peptides when they function. Additionally, some BPs possibly show more than one biofunctionality such as temporin, having antioxidant and antimicrobial activities with a distinctive sequence of amino acids [46]. The unique feature of the BPs of amphibians strongly correlates with their skin histology as a response to the multihabitat, which enable them to face harsh conditions and environments between land and fresh water. Their skins consist of granular and mucus glands, enabling the secretion of several potent BPs as well as protecting them against predators and microbial infections [55]. However, these typical bioactivities may vary after digestion due to the peptide/amino acid sequence modification of BPs. Most of the proteins are completely degraded during gastric digestion, thus generating new BPs with modified sequences [56,57]. Amphibian skin is mainly a tissue featuring multiple functions including respiration, defense, and excretion [26]. Furthermore, this allows them to be superior to other land and marine animals in the production of multifunctional peptides. 

### 3.1. Antioxidant Activity

Antioxidant activity is a limitation of the oxidation of proteins, lipids, DNA, or other molecules that occurs by blocking the propagation stage in oxidative chain reactions. Primary antioxidants directly scavenge free radicals, while secondary antioxidants indirectly prevent the formation of free radicals through Fenton’s reaction [58]. Generally, antioxidants defend against free-radical-induced oxidative damage by various mechanisms. The antioxidant peptides (AOPs) possess scavenging free radicals and reactive oxygen species (ROS) or prevent oxidative damage by interrupting the radical chain reaction of lipid peroxidation [59]. Proteins and their hydrolysates generally possess a significant antioxidant capacity. It is postulated that the capacity comes from their ability to inactivate ROS, quench free radicals, chelate pro-oxidative transition metals, and donate electrons/hydrogen [60]. The size and configuration are reported to be the main factors influencing the antioxidant properties of proteins and their hydrolysates. A higher degree of hydrolysis (DH) will result in a higher production of AOPs. However, the enzyme used to produce AOPs plays an important role in determining the antioxidant potency of the hydrolysates [61]. The ability of the hydrolysate to scavenge radicals varies with the enzyme used, which possibly results from existing differences in the enzyme specificity toward protein substrates. A wide variety of peptides with different modes of action for inhibiting lipid oxidation are generated during hydrolysis. Changes in the size, amount, the exposure of the terminal amino groups of the products obtained, and the composition of free amino acids or small peptides affect the antioxidant activity of peptides [10,62,63]. Several mechanisms involved in the action of AOPs are predominated by scavenging hydroxyl radicals and ROS as well as metal ion chelating.

Peptides are metal chelators, which is one of the antioxidant mechanisms for many BPs. The activity is affected by the size and amino acid sequence [64]. Liu et al. [65] reported that the cleavage of peptides led to an enhanced metal ion binding due to the increased concentration of carboxylic groups and amino groups of the peptides. Carboxyl and amino groups in the side chains of acidic (Glx, Asx) and basic (Lys, His, Arg) amino acids are thought to play an important role in chelating metal ions [66]. Table 2 represents the identified AOPs from various amphibian species, particularly from skin secretions. The antioxidant activities are predominantly evaluated by in vitro assays against DPPH and ABTS, whereas further studies are carried out in silico and in vivo such as cellular oxidative assays [46]. AOPs have also been successfully extracted from the skin of bullfrogs (*R. catesbeiana* Shaw and *R. tigerina*) in the form of collagen hydrolysates. Qian, Jung, and Kim [10] identified APBSP and its sequence from *R. catesbeiana*, which show the half maximal inhibitory concentration (IC_50_) in quenching radicals as follows: DPPH radical (16.1 μM), hydroxyl radical (12.8 μM), superoxide radical (34.0 μM), and peroxyl radical (32.6 μM). Meanwhile, the dominant antioxidant activity of CHs from *R. tigerina* was found under the ABTS radical scavenging activity assay even though they showed other activities against DPPH radical scavenging, ferric reducing antioxidant power (FRAP), ferrous chelating activity (FCA), and oxygen reactive absorbance capacity (ORAC). However, the amino acid sequence has not been identified yet [52]. Some novel AOPs from the skin secretions of four East Asian frog species, namely, daiyunin-1, daiyunin-2, daiyunin-3, maosonensis-1MS1, pleskein-1, pleskein-2, and pleskein-3, were found. Additionally, nigroains (at 50 μM) had 99.7–99.8% and 58.3–68.3% of scavenging activity under the DPPH and ABTS assay, respectively [46]. Most AOPs displayed their potential antioxidant properties (in vitro), which can be extended for further studies on the index of oxidative damage (in vivo) by measuring their ability to increase catalase, superoxide dismutase, and glutathione as well as reduce lactate dehydrogenase [26,67]. Several AOPs have been reported to show antimicrobial activity toward Gram-positive and/or Gram-negative bacteria. This could be correlated with their amino acid sequence, which determines the functional groups and conformation of BPs. Furthermore, these typical BPs could be used as a multifunction supplement in both the food and pharmaceutical industries.

### 3.2. Antimicrobial Activity

Recently, bioactive compounds such as antimicrobial peptides (AMPs) have increasingly been used to counteract the rapidly increasing incidence of bacterial resistance to the usual antibiotics and chemotherapeutics [69]. Each AMP undergoes various mechanisms of action according to its structure and specificity, as shown in Figure 2. The mechanisms of AMPs can be proposed as an aggregate model (Figure 2a), toroidal pore model (Figure 2b), barrel-stave model (Figure 2c), and carpet model (Figure 2d) [70]. Jenssen, Hamill, and Hancock [70] proposed the antimicrobial models performed by BPs that possess an aggregate model with some similarity to the toroidal pore model. However, related molecular studies on specific antimicrobial mechanisms, particularly the BPs from amphibians, have not been reported yet. These mechanisms are only prediction mechanisms proposed by researchers. Positively-charged amino acids in peptide sequences penetrate into bacterial membranes, interacting with the negative charges of the cell wall of phospholipids, promoting cellular lysis [71]. Furthermore, the antibacterial activity of cationic peptides can be modulated through the modification of the net charge or the ratio of hydrophobicity calculated on the percentage of hydrophobic amino acids in the peptide sequence, higher hydrophobicity ratio, and higher antibacterial activity [48,72]. Small peptides have been demonstrated to possess biological activities that are based on their amino acid composition and sequence [73]. However, AMPs from the α-helix group only possessed three main membrane permeation mechanisms: toroidal pore, barrel-stave, and carpet models. However, further details on the mechanism remain unclear [26].

AMPs from amphibian skin and secretions have been reported to exhibit biological activities against the growth of Gram-positive and Gram-negative bacteria as well as fungi. Table 3 represents some examples of AMPs isolated from various species of amphibians. Major AMPs found in amphibian skin secretions have been classified into several groups: esculentin, brevinine, ranatuerin, ranacyclin, temporin, bombinin, and dybowskin [26]. In general, the growth of *Escherichia coli*, *Staphylococcus aureus*, and *Candida albicans* could be inhibited by AMPs at a wide range of minimum inhibitory concentrations (MIC). However, it was not limited to those species of bacteria. Other bacteria were inhibited by AMPs from amphibians such as *Bacterium megaterium, Pseudomonas aeruginosa,* and *Bacillus cereus.* The MIC was affected by molecular diversity in AMPs. It was reported that AMPs could inhibit protein synthesis including DNA and RNA within the bacteria cells [11]. AMPs from Ranid frogs (ranalexin, ranatuerins, gaegurins, brevinins, esculetins, and rugosins) had an intramolecular disulfide bridge that could form a heptapeptide ring with a similar structure at the outside of the cyclic region [74]. This explained the similar antioxidant activity among AMPs. In addition, they might share a similar structural homology related to their ancestors. Previous reviews have hypothesized that most AMPs from amphibians possess the ‘barrel-stave model’ in microbial growth inhibition [26].

### 3.3. Anticancer Activity

Cancer is a disease caused by genetic or epigenetic modifications and cell signaling defects that lead to uncontrollable cell proliferation [87]. Natural products possess an anticancer ability by inducing apoptotic cell death and holding the cell growth in tumor cells with no toxicity to normal cells [88]. Apoptosis possesses two main pathways. The first pathway or death receptor pathway is induced by the CD95 (Fas) death receptor and some other tumor necrosis factor-α (TNF) receptors. The second pathway occurs when the intrinsic or mitochondrial signaling cascade is activated, leading to cytochrome release from the mitochondria, thus forming the complex of apoptosome multiprotein. For instance, cell death can be facilitated by the action of activated caspase-9 at the proteolysis of downstream caspases. These pathways come together in the final activation step of protease cascade, leading to the cleavage of regulatory and structural molecules, and then to cell death [50]. The apoptosis involves some proteases in which proteins (peptides) can act as inhibitors. Table 4 presents the anticancer peptides (ACPs) isolated from various species of amphibians. Brevinins and dermaseptins have been reported to have antimicrobial activities (Table 3), meanwhile, temporins showed antioxidant activity (Table 2). Here, we found that the ACPs had various amino acid sequences and lengths, which varied their bioactivity. 

Brevinine-2R was able to kill cancer cells semi-selectively by activating the novel lysosomal-mitochondrial death pathway (which is not sensitive to caspase inhibitors). However, it induced the overexpression of Bcl2 and inhibited BNIP3, triggering cell death [50]. In vitro/vivo studies were carried out for evaluating dermaseptins as a novel peptide to treat lung cancer. Dermaseptin-PP showed potent antitumor activity with low toxicity and high selectivity and a low propensity to induce resistance [89]. In addition, Temporin-1CEa displayed a low hemolytic effect on human erythrocytes and had no significant cytotoxicity to normal cells, showing a potent antitumor activity. Based on its secondary structure (amphiphilic α-helical cationic peptide), it can be utilized for breast cancer therapy [90]. XLAsp-P1 was isolated and purified from the skin secretions of *Xenopus laevis*, which at the concentration of 5 μg/mL reached up to 55% of inhibition activity against breast cancer cells. Although there was no evidence of the inhibition mechanism, it is possible to cause cell death by deregulating the properties of the phospholipid bilayer, and then inducing membrane leakage [91]. Therefore, ACPs isolated from frog skin can be considered as a new lead agent for anticancer therapy. 

**Table 4 foods-12-01282-t004:** Representative anticancer peptides (ACPs) isolated from various species of amphibians.

Peptides	Species (Common name)	Assay(s): Cell Lines	Amino Acid Sequence	Reference
Brevinin-2R	*Rana ridibunda* (Marsh frog)	MTT assay (in vitro): Jurkat T-cell leukemia, BJAB B-cell lymphoma, MCF-7 breast adenocarcinoma, L929 fibrosarcoma, A549 lung carcinoma.	KLKNFAKGVAQSLLNKASCKLSGQC	[50]
Brevinin-1Da	*Rana dalmatina* (European frog)	–	IIPLLLGKVVCAITKKC	[49]
Brevinin-1RL1	*Rana limnocharis* (Asian rice frog)	MTS assay (in vitro): HCT116 colorectal adenocarcinoma (LD_50_: 5.9 ± 0.2 μM), MDA-MB-231 breast adenocarcinoma (LD_50_: 5.4 ± 0.3 μM), SW480 colorectal adenocarcinoma (LD_50_: 10.4 ± 0.4 μM), A549 lung adenocarcinoma (LD_50_: 5.8 ± 0.2 μM), SMMC-7721 hepatocellular carcinoma (LD_50_: 6.9 ± 0.5 μM), B16-F10 melanomas (LD_50_: 6.6 ± 0.3 μM), NCM460 colon mucosal epithelial (LD_50_: 16.8 ± 0.6 μM), BEAS-2B bronchial epithelial (LD_50_: 16.6 ± 0.3 μM), HaCaT keratinocyte cell (LD_50_: 28.7 ± 0.4 μM).	FFPLIAGLAARFLPKIFCSITKRC	[92]
Dermaseptin-B2	*Phyllomedusa bicolor* (South American tree frog)	Proliferation and angiogenesis assays (in vitro): PC-3 human prostatic adenocarcinoma (EC_50_: 2 μM), NIH-3T3 non-tumor mouse (na.).	GLWSKIKEVGKEAAKAAAKAAGKAAGAVSEAV-CONH_2_	[93]
Dermaseptin-B3	*Phyllomedusa bicolor* (South American tree frog)	Proliferation and angiogenesis assays (in vitro): PC-3 human prostatic adenocarcinoma (EC_50_: 3 μM), NIH-3T3 non-tumor mouse (EC_50_: 0.2–1 µM).	ALWKNMLKGIGKLAGQAALGAVK TLVG-COOH	[93]
Dermaseptin-PP	*Phyllomedusa palliata*	MTT assay (in vitro): H157 human non-small cell lung cancer (IC_50_: 1.6 μM), MCF-7 human breast adenocarcinoma (IC_50_: 2.9 μM), PC-3 human prostate carcinoma (IC_50_: 4.2 μM), U251 MG human neuronal glioblastoma (IC_50_: 2.5 μM). MTT assay (in vivo): H157 tumor in nude mice (at 10^−4^ M induced ~80% LDH release).	ALWKDMLKGIGKLAGKAALGAVKTLV-NH_2_	[89]
Hymenochirin-1B	*Hymennochirus boettgeri* (Congo dwarf clawed frog)	MTT assay (in vitro): HepG2 and PLC human hepatocellular carcinoma cells; NCIH1299, A549 and H460 human lung cancer cells. Cytotoxicity assay (in vitro): A549 lung adenocarcinoma (LD_50_: 2.5 ± 0.2 μM), MDA-MB-231 breast adenocarcinoma (LD_50_: 9.0 ± 0.3 μM), HT-29 colorectal adenocarcinoma (LD_50_: 9.7 ± 0.2 μM), Hep-G2 hepatocarcinoma (LD_50_: 22.5 ± 1.4 μM).	IKLSPETKDNLKKVLKGAIKGAIAVAKMV	[94,95]
Pseudhymenochirin-2Pa	*Pseudhymenochirus merlini* (Merlin’s dwarf gray frog)	Cytotoxicity assay (in vitro): A549 lung adenocarcinoma (LD_50_: 6.0 ± 0.6 μM), MDA-MB-231 breast adenocarcinoma (LD_50_: 6.2 ± 0.6 μM), HT-29 colorectal adenocarcinoma (LD_50_: 11.5 ± 2.6 μM), HUVEC colorectal adenocarcinoma (LD_50_: 68 ± 2 μM).	GIFPIFAKLLGKVIKVASSLISKGRTE	[12]
Pseudhymenochirin-1Pb	*Pseudhymenochirus merlini* (Merlin’s dwarf gray frog)	Cytotoxicity assay (in vitro): A549 lung adenocarcinoma (LD_50_: 2.5 ± 0.2 μM), MDA-MB-231 breast adenocarcinoma (LD_50_: 6.6 ± 0.3 μM), HT-29 colorectal adenocarcinoma (LD_50_: 9.5 ± 1.3 μM), HUVEC colorectal adenocarcinoma (LD_50_: 5.6 ± 0.9 μM).	IKIPSFFRNILKKVGKEAVSLIAGALKQS	[12]
Temporin-1CEa	*Rana chensinensis* (Chinese brown frog)	MTT assay (in vitro): SMMC-7721 human hepatocellular carcinoma (LD_50_: 44.9–50.3 μM), BEL-7402 human hepatocellular carcinoma (LD_50_: 36.9–40.2 μM), Bcap-37 human breast carcinoma (LD_50_: 37.3–39.4 μM), MDA-MB-231 human breast Caucasian adenocarcinoma (LD_50_: 55.0–63.3 μM), MCF-7 human breast carcinoma (LD_50_: 27.8–34.5 μM), LK-2 human squamous cell carcinoma (LD_50_: 58.3–63.4 μM), A-549 human lung adenocarcinoma (LD_50_: 52.1–53.3 μM), NCIH446 human small cell lung carcinoma (LD_50_: 59.4–67.7 μM), BGC-823 human gastric carcinoma (LD_50_: 55.2–63.2 μM), Hela human cervical carcinoma (LD_50_: 36.3–43.9), HO-8910 human ovarian carcinoma (LD_50_: 62.8–66.7 μM), HT-29 human colon carcinoma (LD_50_: >88.91 μM). Hemolysis assay (in vitro): healthy human blood cell (LD_50_: 99.08 μM). LDH assay (in vitro): MCF-7 breast cancer (LD_50_: 40.2–49.1 μM).	FVDLKKIANIINSIF-NH_2_	[90]
XLAsp-P1	*Xenopus laevis* (African clawed frog)	MTT assay (in vitro): MCF-7 breast cancer (LD_50_: <5 μg/mL).	DEDDD	[91]

MTS: 3-(4,5-dimethylthiazol-2-yl)-5-(3-carboxymethoxyphenyl)-2-(4-sulfophenyl)-2H-tetrazolium; MTT: 3-(4,5-dimethylthiazol-2-yl)-2,5-diphenyl-2H-tetrazolium bromide; LDH: lactate dehydrogenase; LD_50_: concentration equal to produce ~50% cell death; EC_50_: median effective concentration; IC_50_: maximal inhibitory concentrations.

### 3.4. Antidiabetic Activity

Type 2 DM is caused by a progressive imbalance between diet and physical activity (obesity), leading to insulin resistance, with a decrease in production level underlying the disease [96]. Insulin is the main regulating hormone in the development of type 2 DM in which it regulates blood sugar levels in the body by allowing glucose to enter the body’s cells [97]. Nevertheless, other glucose-regulating enzymes have been presumed to evaluate the antidiabetic activity of BPs. The antidiabetic mechanisms of action were proposed by Soltaninejad et al. [98] and are presented in Figure 3. Although some particular ADP mechanisms have not been identified, an ADP may exhibit multiple mechanisms of action. This agrees with their overlapping biofunctionalities, which occur in the same isolated ADP. For instance, tigerinins, BPs from *R. tigerina*, have been studied for their antimicrobial, wound healing, and antidiabetic properties [5,14,24]. Several peptides from frog skin were identified to stimulate insulin release both in vitro and in vivo as well as inhibit the regulatory enzymes in blood glucose levels such as α-glucosidase [24,51,98,99]. Some representative antidiabetic peptides (ADPs) from amphibians are shown in Table 5. Bombesins, brevinins, dermaseptins, esculetins, magainins, ranateurins, and temporins were the major ADPs found in edible frog species. Brevinins, esculentins, and RK-13 involve the cyclic AMP-protein kinase A- and C-dependent G-sensitive pathways, affecting the Rho G proteins. Temporins, hymenochirin-1B, and pseudhymenochirins play a role in the KATP channel-independent pathway. Ranateurins can retain the plasma membrane integrity. Bombesins, GM-14, and IN-21 show antidiabetic activity in the G protein-insensitive pathway by involving cAMP-dependent. Phylloseptin-L2 involves an influx of Ca^2+^ or the closure of ATP-sensitive K^+^ channels. Pseudin-2 plays a role in Ca^2+^-independent pathways. Alyteserin-2a and magainins depolarize and increase the intracellular Ca^2+^ concentration. Ocellatin-L2, plasticin-L1, pseudin-2, tigerinins, palustrins, xenopsins, dermaseptins, and caerulin-B1 stimulate insulin release from BRIN-BD111.

In addition, glucagon-like peptide 1 (GLP-1) was stimulated by BPs such as tigerinin-1R, magainin-AM1, -AM2, CPF-AM1, and PGla-AM1. Tigerinin-1R improved the glucose homeostasis and beta cell function in mice [24]. An in vivo study found that CPF-AM1 regulated both direct and indirect insulinotropic action and secretion [14,51]. Other than skin secretion, a novel peptide from the collagen hydrolysate from the skin of CGS, namely, GPPGPA, showed an inhibitory effect against α-glucosidase and a protective effect on insulin resistance and the oxidative stress of IR-HepG2 cells [99]. The molecular docking study predicted that GPPGPA was bound to the core targets (AKT1, MAPK8, MAPK10, and JUN) via a ‘peptide–target-pathway’ network. Thus, it enabled GPPGPA to be utilized as a hyperglycemic inhibitor in functional food. Some novel ADPs have not been identified for their antidiabetic mechanisms, but might possess the multiple-mechanisms above-mentioned in Figure 3. For instance, the possible mechanism for insulin stimulation by dermaseptin-B4 and -LI1 has not been determined [98]. This can be a new challenge for further research to evaluate the unidentified antidiabetic activity of ADPs.

### 3.5. Other Bioactivities

The host-defense peptides of amphibians also display some other biofunctional features including antiviral, wound healing, and immunomodulatory activities [82,113]. Brevinin-1 and ranateurins featured the antiherpes virus with strong hemolytic activity [26]. A potential wound healing-promoting peptide (AH90, ATAWDFGPHGLLPIRPIRIRPLCG) was identified from the frog skin of *Odorrana graham* that helped to promote endogenous wound healing agents (transforming growth factor β1) without mitogenic effects [13]. Wound healing is a continuous process and consists of several complex mechanisms. This can be simplified into four main phases: (1) coagulation and hemostasis phase; (2) inflammatory phase; (3) proliferative phase; and (4) remodeling phase [114,115]. In this case, this typical BP is involved in the proliferative and remodeling phases. A frog skin powder from *R. ridibunda* was formulated that showed significant healing and antibacterial effects on wounds [2]. In addition, the skin secretions of *R. ridibunda* exhibited a promotion in the wound healing process. This feature is strongly related to the ability of amphibians to overcome frequent tissue injuries. Amphibians from urodele (i.e., salamanders) commonly perform a higher rate of healing than those from anuran (frogs and toads) [13]. Cathelicidin-OA1 (IGRDPTWSHLAASCLKCIFDDLPKTHN) was found as a potent skin wound healing accelerator isolated from *Odorrana andersonii* [47]. Pseudhymenochirin-1Pb and -2Pa were reported to play an immunoregulatory role [12]. CGS skin was formulated with tung oil and used in the therapy of burns and is considered as a folk prescription [7]. Nevertheless, collagens have been extracted from frog and salamander skins, referring to their future feasibility in tissue engineering [44,53,116,117,118,119,120,121,122]. Based on the functionalities of BPs from amphibians, it is highly possible to utilize them in the food and pharmaceutical industries as excellent multifunctional ingredients or even medicine. 

## 4. Multiple Constraints toward Amphibian Skin Derivatives

Although amphibians offer an opportunity to produce BPs with excellent functionalities and health benefits, some essential aspects should be addressed before their utilization in the food and biomedical industries. Several constraints raised in this review are related to (1) food safety and health; (2) culture and religion; (3) ecological and animal welfare; and (4) sustainable production [30,33,38,123,124]. Thus, a cross-sectoral transdisciplinary approach is required to evaluate the exploitation of the meat and skin of amphibians. In terms of eating quality, amphibian skin has particular and unpleasant odorants due to the occurrence of enzymatic reactions, lipid oxidations, and microbial activities in the skin [116,125]. For instance, the skin of CGS has earthy, muddy, medicinal, rancid, and musty odorants. The odorant profile of collagen extracted from CGS skin varied depending on the extraction methods used. The acid-soluble collagen perceived sour, ammonia-like, and acrid off-odors while the pepsin-soluble collagen had a low intensity of off-odors [116]. The disgust perception among humans toward amphibians was evaluated in which anurans were less disgusting than caecilians that were worm-like, legless, and small-eyed. Fears and phobias of the animal were presumed to be involved in the recent perception [126]. Mathew, Ndamitso, Yanda Shaba, Mohammed, Salihu, and Abu [38] reported a trace amount of antinutritive compound found in *P. esculentus* (i.e., tannins and oxalates) that was relatively lower than that of other meat sources. Essentially, tannins and oxalates could be complexed with bivalent mineral ions, thus affecting the bioavailability of composite nutrients. Moreover, diabetic people should take precautions before consuming frog meat due to its high content of minerals with a ratio of Na/K above one. Another food safety concern is the foodborne diseases caused by zoonotic pathogens and antimicrobial-resistant bacteria that exist in amphibian food products. The illegal trade and improper manufacturing practices (i.e., cross-contamination, poor hygiene during handling, processing, and storage) contribute to most cases that occur in the European Union [35,123]. In the last three years, global outbreaks of coronavirus (COVID-19) occurred for the first time in Beijing, China, then spread widely all over the world. It was caused by the spread of severe acute respiratory syndrome coronavirus 2 (SARS-CoV-2). This virus was reported as related to the fresh meat and seafood industries, and air-transmitted rapidly as a foodborne pathogen. SARS-CoV-2 can contaminate the food chain of an infected animal, starting from slaughtering, trimming, and packaging or even associated environments [127]. Due to the chemical defense ability of amphibians, the chemical nature, origin, and function of noxious substances are necessary to evaluate. A little information about the origin and function of poisons and noxious substances found in amphibians is known. Biogenic amines (i.e., serotonin, histamine, and tyramine) are found in the skin of various toads and frogs, and water-soluble alkaloid tetrodotoxin and lipophilic alkaloids are found in salamanders and some frog species. Some amphibian skin alkaloids are sequestered from the diet, which are from small arthropods. Nevertheless, under the proper farming and BP isolation procedures, these non-targeted compounds can be eliminated [128]. 

Strong local wisdom may still exist in China where it is believed that touching a CGS can bring bad luck and is taboo. Moreover, it sounds like a human baby’s cry and is considered ‘dirty’ [30]. This might result in more limitations to the exploitation of CGS. For Muslims, the consumption and/or utilization of amphibians are not permissible at all and are known as haram. This practice has been forbidden based on a hadith narrated by prophets. Nevertheless, no specific rule has been reported on the status of amphibians in the Qur’an. Thus, it is probably the only madhab that permits the consumption and utilization of amphibians, particularly frogs. However, modern Muslim scientists and jurists are openly questioning and discussing this also in regard to the current status of amphibians for medical and research purposes [55]. Several frogs and salamanders are listed in the IUCN Red List of Threatened Species. It is suggested to recheck the species before use for consumption and/or trade. Overexploitation of amphibians can consequently influence the ecosystem structure and function [33]. A decline in the frog population is projected to give an impact on human health, as vector-borne diseases carried by mosquitoes can increase and treat humans. A dramatical reduction in tadpoles will increase the algal biomass by affecting the algal assemblage structure, thus the organic and inorganic sediments will be accumulated. Moreover, the water quality will then be altered [129]. Nevertheless, the captive breeding of edible amphibians has been addressed to tackle these ecological and sustainability issues [30,130]. Hunting, trade, and cultural use are identified as the greatest threats to diversity and availability [1]. The exploration of amphibians will respect ethical food choices without harming and disrupting food production. This practice can be considered to respect animal welfare, thus eluding their extinction [6,124]. Captive breeding and conservation need to be executed at the same time in order to sustain the amphibian population [30,130,131]. Therefore, a multi-sectoral approach has to be evaluated prior to up-scaling the production of amphibians.

## 5. Prospective Application in Food and Biomedical Industries

### 5.1. Application in Food Industry

Amphibian skins are regarded as an agricultural by-product that is rich in proteins and lipids. As a collagenous source, it can be utilized to produce collagen, gelatin, and its hydrolysates. In general, gelatin and its hydrolysates have been utilized in food products as functional food additives or supplements. Focusing on their features, BPs have been developed and formulated as food preservatives and as bioactive components of edible films and coatings [132]. Moreover, the application of collagen hydrolysate (CH) in the biodegradable film could effectively give more added value to the resulting CH in the food supply chain, considering the sustainability aspect. Protein hydrolysates containing BPs are used as components of biopolymer films not only to improve the mechanical properties of the film packaging, but also primarily as active ingredients extending the shelf-life of packaged foods [132]. The presence of BPs in protein hydrolysates makes it a functional ingredient that is applied in the food and medicine industries [133]. For instance, gelatin extracted from Asian bullfrog skin showed a similar gel strength to bovine gelatin [25,54]. Frog skin gelatin can be used as a biocompatible material, which is safe for human consumption based on cytotoxicity studies [25]. Gelatin from the skin of Chinese giant salamanders (CGS) exhibited good interfacial properties by enhancing foam expansion and foam stability. Thus, it has the potential to be used as a novel ingredient in food emulsion systems [45]. CHs have shown some bioactivities such as antioxidant and antimicrobial activities, enabling them to be utilized as a daily food supplement. Furthermore, the regular consumption of CH aims to enhance the production or regeneration of collagen in the human body [134]. Prior to the consumption of BPs, it is essential to consider the bioaccessibility of BPs during and after digestion, where the peptide modifications occurred (i.e., degradation and hydrolysis). This will generate smaller peptides with various sequences due to their bioactivities [57]. However, it was presumed that proline-containing peptides (such as CH) have a high resistance to hydrolysis by the digestive enzymes [56].

### 5.2. Application in Biomedical Industry 

In the biomedical industry, amphibian skins and secretions have more potential for application. Nevertheless, collagen extracted from the skin of frogs and CGSs can be an alternative source of tissue engineering agents. Collagen can accelerate wound healing rapidly and effectively, which enhances immunomodulation [135,136]. This implies the functional property of collagen as a biomaterial scaffold. In addition, an in vivo study of alternative wound dressings using collagen, particularly from marine sources, showed a significant rapid wound healing mechanism in a rat model [135]. Nowadays, collagens have been utilized widely not only in the food industry, but also in the pharmaceutical industry due to their various bioactivities [137,138]. Another alternative source of collagen, especially from industrial by-products, would be potential and promising research in preparing wound dressings. The BPs of different frog skins possessed antimicrobial, anticancer, antidiabetic, immunomodulatory, and many more activities [12]. This also offers another application of BPs as a multifunctional and healthy supplement for particular consumer groups (i.e., elderly and /or patients with chronic disease). In addition, it suggests future cellular studies using AOPs in topical formulations for antiaging pharmaceuticals, or as putative neuroprotectors, which may assist in the development of therapeutic strategies aiming at controlling oxidative stress in neurological diseases [27]. BPs were evaluated on their anticancer activity, thus broadening their possibility as future cancer treatment agents. Nevertheless, BPs from amphibians can be formulated with other bioactive compounds to develop novel drugs or act as drug-delivery agents. 

## 6. Conclusions

Amphibian skin and secretion have been explored as an alternative source of collagen and bioactive peptides (BPs). They provide novel BPs with various excellent bioactivities. The isolated peptides feature at least one of the following bioactivities: antioxidant, antimicrobial, anticancer, antidiabetic, anti-inflammatory, antitumor, antiviral, etc. The amino acid sequence and length were unique and distinctive due to different amphibian species. Some pros and cons have been addressed regarding the consumption of amphibians and their derivative products. Although there is a great opportunity to utilize this BP source, the exploitation should be controlled to avoid amphibian extinction and indirectly conserve the ecological system. Therefore, amphibians, particularly frogs and giant salamanders, can be utilized as a feasible source to improve human health and well-being with proper practices.

## Figures and Tables

**Figure 1 foods-12-01282-f001:**
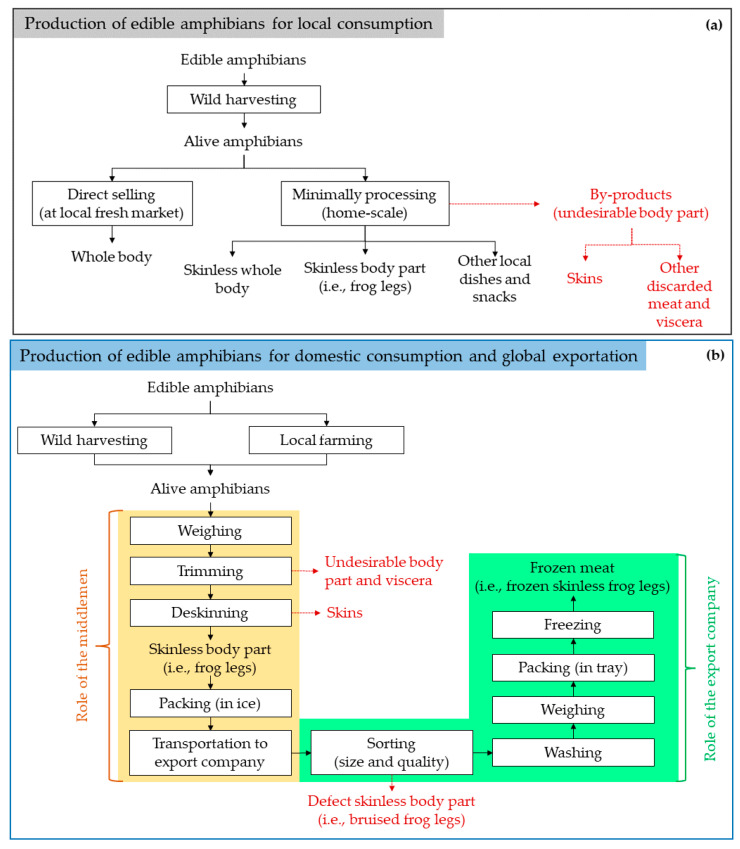
General production schemes of edible amphibians for (**a**) local consumption and (**b**) domestic consumption and global exportation, adapted from Kusrini and Alford [34] and Cunningham, Turvey, Zhou, Meredith, Guan, Liu, Sun, Wang, and Wu [30].

**Figure 2 foods-12-01282-f002:**
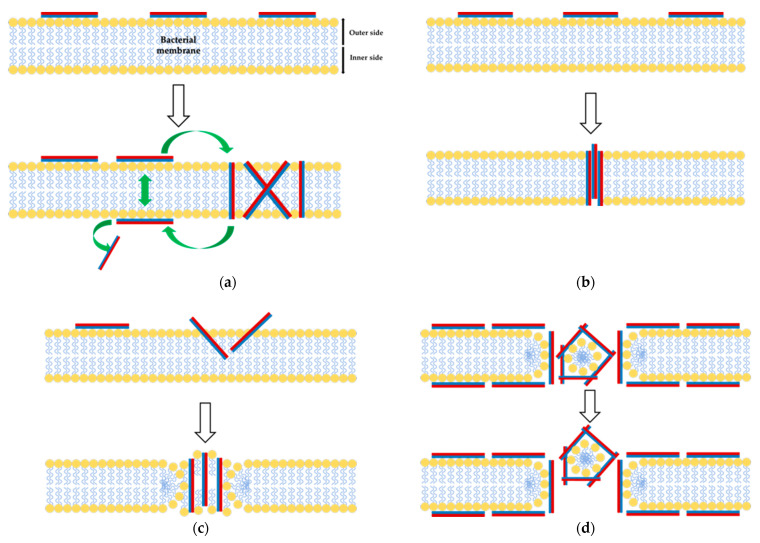
Mechanisms of action of antimicrobial peptides (the peptides are shown as cylinder rods, where the hydrophilic regions are presented in red color and the hydrophobic regions in blue). (**a**) Aggregate model; (**b**) toroidal pore model; (**c**) barrel-stave model; (**d**) carpet model. Adapted with permission from ref. [70]. Copyright © 2023 American Society for Microbiology.

**Figure 3 foods-12-01282-f003:**
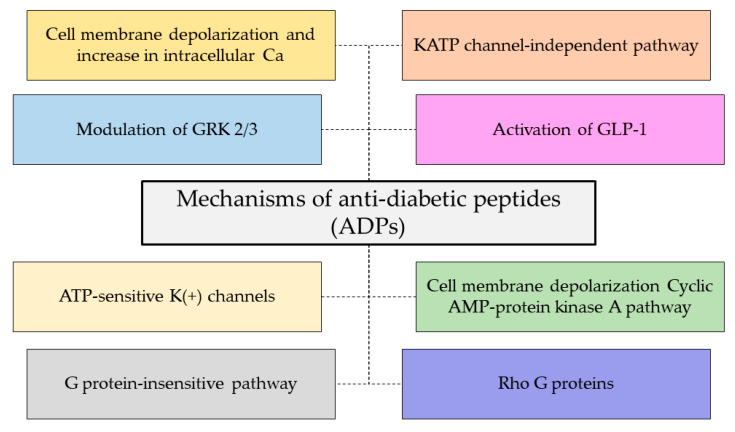
Mechanisms of action of antidiabetic peptides [98].

**Table 1 foods-12-01282-t001:** Nutritional composition of common species of edible amphibians.

Scientific Name (Common Name)	Body Part	Moisture (g/100 g)	Protein (g/100 g)	Fat (g/100 g)	Carbohydrate (g/100 g)	Ash (g/100 g)	Reference
**Frogs**							
*Bufo terrestris* (Southern toads)	Whole body	68.9–74.8	16.8–17.5	2.8–5.1	nd.	3.2–3.8	[39]
*Hoplobatrachus occipitalis* (Crowned bullfrog)	Meat	71.7–77.8	16.9–19.5	1.1–1.8	4.6	5.0	[40]
*Pelophylax esculentus* (European frog)	Meat and bones	77.7	7.0	3.6	6.5	2.0	[38]
*Pyxicephalus adspersus* (Giant African bullfrog)	Thigh and calf	71.9–78.5	6.4–12.0	1.7–3.7	0.5–0.6	4.0–6.0	[41]
*Rana catesbiana* (American bullfrog)	Fore-chest	83.4	15.3	0.6	nd.	0.6	
	Thigh	76.9	21.2	1.4	nd.	0.8	[42]
	Calf	78.4	17.7	1.0	nd.	0.8	
*Rana clamitans/Lithobates clamitans* (Green frog/wood frog)	Whole body	76.2–78.8	15.3–16.7	1.5–3.2	nd.	2.7–3.3	[39]
*Rana esculenta* (Green frog)	Fore-chest	79.5	18.9	1.2	nd.	0.6	[43]
	Thigh and calf	79.7	19.2	0.7	nd.	0.6	
*Rana nigromaculata* (Black-spotted frog)	Skin	74.0	6.3	0.1	19.1	0.5	[44]
*Rana ridibunda* (Marsh frog)	Meat	79.8–82.8	10.5–15.7	0.7–1.5	0.9–1.3	nd.	[37]
**Salamanders**							
*Andrias davidianus* (Chinese giant alamander)	Meat	79.0–82.3	14.0–16.4	2.5–3.5	nd.	0.7–1.1	[7]
	Skin	67.6	29.1	1.21	nd.	0.5	[45]

nd.: not detected.

**Table 2 foods-12-01282-t002:** Representative antioxidant peptides (AOPs) isolated from various species of amphibians.

Peptides	Species (Common Name)	Assay(s)	Amino Acid Sequence	Reference
Andersonin-C1	*Odorrana margaretae* (Green odorous frog)	–	TSRCIFYRRKKCS	[26]
Andersonin-G1	*Odorrana andersonii* (Golden crossband frog)	–	KEKLKLKAKAPKCYNDKLACT	[26]
Andersonin-H3	*Odorrana margaretae* (Green odorous frog)	–	VAIYGRDDRSDVCRQVQHNWLVCDTY	[26]
Antioxidin-RP1	*Rana pleuraden* (Yunnan pond frog)	ABTS (at 80 μg/mL of peptide): 95.9 ± 4.7% *, DPPH (at 80 μg/mL of peptide): 96.9 ± 5.1% *, NO (at 80 μg/mL of peptide): 66.4 ± 9.8% *, FRAP (at 80 μg/mL of peptide, 700 nm): 0.41 ± 0.007.	AMRLTYNKPCLYGT	[68]
APBSP	*Rana catesbeiana Shaw* (American bullfrog)	DPPH (at 1.5 mg/mL of peptide): 16.6–58.6% *, Hydroxyl radical (at 1.5 mg/mL of peptide): 32.8–75.2% *, Superoxide radical (at 1.5 mg/mL of peptide): 10.9–51.5% *, Peroxyl radical (at 1.5 mg/mL of peptide): 36.0–59.7% *.	LEELEEELEGCE	[10]
Cathelicid-OA1	*Odorran andersonii* (Golden crossband frog)	ABTS (at 4–32 μM of peptide): ~12.5–90% *, DPPH (at 128 μM of peptide): ~90% *.	IGRDPTWSHLAASCLKCIFDDLPKTHN	[47]
Nigroain-B-MS1	*Hylarana maosuoensis* (Maoson frog)	ABTS (at 50 μM of peptide): 68.3% *, DPPH (at 50 μM of peptide): 99.7% *.	CVVSSGWKWNYKIRCKLTGNC	[46]
Nigroain-C-MS1	*Hylarana maosuoensis* (Maoson frog)	ABTS (at 50 μM of peptide): 58.3% *, DPPH (at 50 μM of peptide): 99.8% *.	FKTWKNRPILSSCSGIIKG	[46]
OA-VII2	*Odorrana andersonii* (Golden crossband frog)	Index of oxidative damage in vivo (HaCaT cells exposed with UVB irradiation mouse skin, at 0.5–10 μM of peptide): increase of CAT (23.9–54.5%), reduction of LDH (73.5–100.7%), increase of SOD (79.4 ± 2.0%), increase of GSH (67.4 ± 10.1%).	VIPFLACRPLGL	[26,67]
OA-GL21	*Odorrana andersonii* (Golden crossband frog)	ABTS (at 500 µM of peptide): ~70% *, DPPH (at 500 µM of peptide): <10% *.	GLLSGHYGRVVSTQSGHYGRG	[26]
OM-LV20	*Odorrana margaretae* (Green odorous frog)	ABTS: na., DPPH: na., NO (at 1 mM of peptide): 40.1 ± 5.8% *.	LVGKLLKGAVGDVCGLLPIC	[26]
OS-LL11	*Odorrana schmackeri* (Schmacker’s frog)	ABTS (at 1 μM of peptide): 20–30% *, DPPH (at 1 μM of peptide): <10% *.	LLPPWLCPRNK	[26]
Pleurain-A1	*Rana pleuraden* (Yunnan pond frog)	ABTS (at 80 μg/mL of peptide): 74.8 ± 2.2% *, DPPH (at 80 μg/mL of peptide): na., NO (at 80 μg/mL of peptide): 21.1 ± 5.2% *, FCA (at 80 μg/mL of peptide, 700 nm): 0.15 ± 0.04.	SIITMTKEAKLPQLWKQIACRLYNTC	[26,68]
Pleurain-D4	*Rana pleuraden* (Yunnan pond frog)	ABTS (at 80 μg/mL of peptide): na., DPPH (at 80 μg/mL of peptide): 30.3 ± 12.7% *, NO (at 80 μg/mL of peptide): 35.7 ± 3.6% *, FRAP (at 80 μg/mL of peptide, 700 nm): 0.21 ± 0.02.	FLSGILKLAFKIPSVLCAVLKNC	[26,68]
Pleurain-E1	*Rana pleuraden* (Yunnan pond frog)	ABTS (at 80 μg/mL of peptide): 6.3 ± 8.5% *, DPPH (at 80 μg/mL of peptide): 69.1 ± 9.6% *, NO (at 80 μg/mL of peptide): 11.5 ± 2.9% *, FRAP (at 80 μg/mL of peptide, 700 nm): 0.29 ± 0.02.	AKAWGIPPHVIPQIVPVRIRPLCGNV	[26,68]
Ranacyclin-HB1	*Pelophylax hubeiensis* (Hubei gold-striped pond frog)	ABTS (at 50 μM of peptide): 49.3 ± 3.8% *, DPPH (at 50 μM of peptide): 11.6 ± 2.4% *.	GAPKGCWTKSYPPQPCFGKK	[46]
Salamandrin	*Salamandra salamandra* (Fire salamander)	ABTS: 0.3 ± 0.0 Trolox equivalent/mg of peptide, DPPH: 0.1 ± 0.0 Trolox equivalent/mg of peptide).	FAVWGCADYRGY	[27]
Temporin-MS1	*Hylarana maosuoensis* (Maoson frog)	ABTS (at 50 μM of peptide): 21.4 ± 2.2% *, DPPH: na.	FLTGLIGGLMKALGK	[46]

* Indicates the percent inhibition or scavenging capacity of radicals used in the assay. ABTS: 2,2′-azino-bis (3-ethylbenzothiazoline-6-sulfonic acid); DPPH: 2,2-diphenyl-1-picrylhydrazyl; NO: nitric oxide; FRAP: ferric reducing antioxidant power; CAT: catalase; SOD: superoxide dismutase; GSH: glutathione; LDH: lactate dehydrogenase; na.: no activity; –: not specified.

**Table 3 foods-12-01282-t003:** Representative antimicrobial peptides (AMPs) isolated from various species of amphibians.

Peptides	Species (Common Name)	MIC against Microorganisms	Amino Acid Sequence	Reference
Brevinin-1	*Rana brevipoda* (Daruma pond frog)	*Staphylococcus aureus* (8–137.5 μM)*, Enterococcus faecalis* (37.5 μM)*, Nocardia asteroids* (37.5 μM)*, Escherichia coli* (34–37.5 μM).	FLPVLAGIAAKVVPALFCKITKKC	[5,46]
Brevinin-1Da	*Rana dalmatina* (European frog)	*S. aureus* (7 μM), *E. coli* (30 μM).	IIPLLLGKVVCAITKKC	[49]
Brevinin-1E	*Rana esculenta* (Green frog)	*S. aureus* (0.6 μM), *E. coli* (1.8 μM).	FLPLLAGLAANFLPKIFC_1_KITRKC_1_	[75,76]
Brevinin-1T	*Rana temporaria* (Common frog)	–	VNPIILGVLPKFVCLITKKC	[49]
Brevinin-2RNa	*Rana nigromaculata* (Black-spotted frog)	–	GLFDVVKGVLKGVGKNVAGSLLEQLKCKLSGGC	[77]
Buforin-2	*Bufo bufo garagrizans* (Asian toad)	*Bacillus subtilis* (2 μg/mL), *S. aureus* (4 μg/mL), *Streptococcus mutans* (2 μg/mL), *Streptococcus pneumoniae* (4 μg/mL), *E. coli* (4 μg/mL), *Salmonella typhimurium* (1 μg/mL), *Serratia marcescens* (4 μg/mL), *Pseudomonas putida* (2 μg/mL), *Candida albicans* (1 μg/mL), *Cryptococcus neoformans* (1 μg/mL), *S. cerevisiae* (1 μg/mL).	TRSSRAGLQFPVGRVHRLLRK	[78]
Dermaseptin-S1	*Phylomedusa sauvagil* (Hylinae tree frog)	*S. aureus* (12 μM), *E. coli* (12 μM), *Pseudomonas aeruginosa* (>24 μM), *C. albicans* (>24 μM).	ALWKTMLKKLGTMALHAGKAALGAAADTISQGTQ	[5,21,79]
Esculentin-1	*Rana esculenta* (Green frog)	*E. coli* (0.2 μM), *Bacillus meganterium* (0.1 μM), *S. aureus* (0.4 μM), *P. aeruginosa* (0.7 μM), *C. albican* (0.5 μM), *S. cerevisiae* (0.9 μM).	GIFSKLGRKKIKNLLISGLKNVGKEVGMDVVRTGIDAGOKIKGEC	[5,26]
Gaegurin-5	*Rana rugosa* (Wrinkled frog)	–	FLGALFKVASKVLPSVKCA*Im*TKKC	[5]
Japonicin-2	*Rana chaochiaoensis* Liu (Chinese brown frog)	*S. aureus* (20 μM), *E. coli* (12 μM).	FGLPMLSILPKALCILLKRKC	[80]
Magainin-2	*Xenopus laevis* (African clawed frog)	*C. albicans, Trichosporon beigelii, Aspergillus flavus, Fusarium oxyspovrum* (ns.).	GIGKFLHSAKKFGKAFVGEIMNS	[5,75]
Maximin-H5	*Bombina maxima* (Chinese red belly toad)	*S. aureus* (90 µM).	ILGPVLGLVSDTLDOVLGIL-NH_2_	[81]
Nigrocin-1	*Rana nigromaculata* (Black-spotted frog)	*Micrococcus luteus* (2.5 μg/mL), *Shigella dysenteriae* (10 μg/mL), *Klebsiella pneumoniae* (10 μg/mL), *P. aeruginosa* (75 μg/mL), *S. typhimurium* (22.5 μg/mL), *Proteus mirabilis* (>200 μg/mL), *S. marcescens* (>200 μg/mL), *C. albicans* (100 μg/mL).	GLLDSIKGMAISAGKGALQNLLKVASC_1_KLDKTC_2_	[77]
Nigrocin-2	*Rana nigromaculata* (Black-spotted frog)	*M. luteus* (2.5 μg/mL), *S. dysenteriae* (10 μg/mL), *K. pneumoniae* (10 μg/mL), *P. aeruginosa* (100 μg/mL), *S. typhimurium* (22.5 μg/mL), *P. mirabilis* (>200 μg/mL), *S. marcescens* (>200 μg/mL), *C. albicans* (150 μg/mL).	GLLSKVLGVGKKVLC_1_GVSGLC_2_	[77]
Pseudhymenochirin-1Pb	*Pseudhymenochirus merlini* (Merlin’s dwarf gray/clawed frog)	*E. coli* (10 μM), *K. pneumoniae* (20 μM), *P. aeruginosa* (20 μM), *C. albicans* (80 μM).	IKIPSFFRNILKKVGKEAVSLIAGALKQS	[12]
Pseudhymenochirin-2Pa	*Pseudhymenochirus merlini* (Merlin’s dwarf gray/clawed frog)	*E. coli* (>80 μM), *K. pneumoniae* (>80 μM), *P. aeruginosa* (>80 μM), *C. albicans* (80 μM).	GIFPIFAKLLGKVIKVASSLISKGRTE	[12]
Ranacyclin-B-RN1	*Hylarana* *Nigrovittata* (Black-striped frog)	*S. aureus* (6 μM).	SALVGCWTKSYPPKPCFGR	[82]
Ranacyclin-B-RN1	*Hylarana* *Nigrovittata* (Black-striped frog)	*S. aureus* (12.7 μM).	SALVGCGTKSYPPKPCFGR	[82]
Ranacyclin-E	*Rana temporaria* (Common frog)	Lethal concentration: *E. coli* (na., >100 μM), *Yersinia pseudotuberculosis* (9 μM), *Pseudomonas syringae* pv *tabaci* (80 μM), *B. megaterium* (3 μM), *Staphylococcus lentus* (7 μM), *M. luteus* (5 μM), *C. albicans* (na., >100 μM), *Candida tropicalis* (7.4 μM), *Candida guillier-mondii* (3.4 μM). MIC: *Phytophthora nicotianae* spores (32 μM).	SAPRGCWTKSYPPKPCK	[83]
Ranacyclin-T	*Rana esculenta* (Green frog)	Lethal concentration: *E. coli* (30 μM), *Y. pseudotuberculosis* (5 μM), *Ps. syringae* pv *tabaci* (16 μM), *B. megaterium* (3 μM), *S. lentus* (10 μM), *M. luteus* (8 μM), *C. albicans* (22 μM), *C. tropicalis* (14 μM), *C. guillier-mondii* (1.0 μM). MIC: *P. nicotianae* spores (16 μM).	GALRGCWTKSYPPKPCK	[83]
Ranacyclin-NF	*Pelophylax nigromaculatus* (Black-spotted frog)	*S. aureus* (512 μM), *E. faecalis* (>512 μM), *E. coli* (>512 μM), *P. aeruginosa* (>512 μM), *P. pneumoniae* (>512 μM), *C. albicans* (>512 μM).	GAPRGCWTKSYPPQPCF	[84]
Ranacyclin-NF3L	*Pelophylax nigromaculatus* (Black-spotted frog)	*S. aureus* (>512 μM), *E. faecalis* (>512 μM), *E. coli* (>512 μM), *P. aeruginosa* (>512 μM), *P. pneumoniae* (>512 μM), *C. albicans* (>512 μM).	GALRGCWTKSYPPQPCF	[84]
Ranalexin	*Rana catesbeiana* (American bullfrog)	*S. aureus* (4 μg/mL), *E. coli* (32 μg/mL), *P. aeruginosa* (128 μg/mL).	FLGLIKIVPAMIC_1_AVTKKC_1_	[11,75]
Ranateurin-4	*Rana temporaria* (Common frog)	–	FLPFIARLAAKVFPSIICSVTKKC	[5]
Ranateurin-T	*Rana temporaria* (Common frog)	*S. aureus* (120 μM), *E. coli* (40 μM), *C. albicans* (na., >150 μM).	GLLSGLKKVGKHVAKNVAVSLMDSLKCKISGDC	[74]
Temporin-A	*Rana temporaria* (Common frog)	*S. aureus* (2.6–5.2 μM), *E. faecalis* (20.9 μM), *E. faecium* (10.5 μM).	FLPLIGRVLSGIL-*Am*	[5,85]
Temporin-B	*Rana temporaria* (Common frog)	*K. pneumoniae* (128 μg/mL), *A. baumanii* (32–64 μg/mL)*, P. aeruginosa* (>128 μg/mL)*, E. coli* (64 μg/mL)*, S. aureus* (16–32 μg/mL)*, E. faecalis* (64 μg/mL)*, C. albicans* (32 μg/mL).	LLPIVGNLLKSLL-*Am*	[75]
Temporin-L	*Rana temporaria* (Common frog)	*K. pneumoniae* (16 μg/mL), *A. baumanii* (4 μg/mL)*, P. aeruginosa* (16–64 μg/mL)*, E. coli* (4 μg/mL)*, S. aureus* (2–4 μg/mL)*, E. faecalis* (4–8 μg/mL)*, C. albicans* (8 μg/mL).	FVQWFSKFLGRIL	[75]
Temporin-1RNa	*Rana nigromaculata* (Black-spotted frog)	*E. coli* (25 μM), *P. aeruginosa* (25 μM)*, S. aureus* (12.5 μM)*, B. cereus* (6.25 μM)*, Streptococcus lactis* (6.25 μM)*, C. albicans* (12.5 μM).	ILPIRSLIKKLL-NH_2_	[44,86]
Temporin-1RNb	*Rana nigromaculata* (Black-spotted frog)	*E. coli* (12.5 μM), *P. aeruginosa* (12.5 μM)*, S. aureus* (3.13 μM)*, B. cereus* (3.13 μM)*, S. lactis* (3.13 μM)*, C. albicans* (6.25 μM).	FLPLKKLRFGLL-NH_2_	[78,86]
Tigerinin-1	*Rana tigerina* (Asian bullfrog)	*B. subtilis* (30 μg/mL), *S. aureus* (30 μg/mL), *E.coli* (40 μg/mL), *P. putida* (40 μg/mL), *M. luteus* (20 μg/mL), *S. cerevisiae* (80 μg/mL).	FC_1_TMIPIPRC_2_Y-*Am*	[5]
Tigerinin-2	*Rana tigerina* (Asian bullfrog)	*B. subtilis* (20 μg/mL), *S. aureus* (40 μg/mL), *E.coli* (50 μg/mL), *P. putida* (50 μg/mL), *M. luteus* (20 μg/mL), *S. cerevisiae* (100 μg/mL).	RVCFAIPLPICH-*Am*	[5]
Tigerinin-3	*Rana tigerina* (Asian bullfrog)	*B. subtilis* (30 μg/mL), *S. aureus* (30 μg/mL), *E.coli* (40 μg/mL), *P. putida* (40 μg/mL), *M. luteus* (30 μg/mL), *S. cerevisiae* (80 μg/mL).	RVCYAIPLPICY-*Am*	[71]

MIC: Minimum inhibitory concentrations; *Am*: C-terminal amidate; na.: not active; –: not specified.

**Table 5 foods-12-01282-t005:** Representative antidiabetic peptides (ADPs) isolated from various species of amphibians.

Peptides	Species (Common Name)	Assay(s)	Amino Acid Sequence	Reference
Alyteserin-2a	*Alytes obstetricans* (Midwife toad)	Insulin-releasing activity (in vitro): BRIN-BD11 clonal β-cells (TC: 30 nM, Basal release of insulin at 3 μM: 296 ± 26%). *In vivo* studies: ns.	ILGKLLSTAAGLLSNL	[100]
Amolopin	*Amolops loloensis* (Loloku sucker frog)	Insulin release (in vitro): INS-1 (significantly increased insulin release from 40 to 50 mLU/L at 12.5–50 μg/mL).	FLPIVGKSLSGLSGKL	[98]
Bombesin (protein fractions)	*Bombina variegate* (Yellow-bellied toad)	Insulin-releasing activity (in vitro): BRIN-BD11 clonal β-cells (insulin secretion: 1.5–3.5 ng/10^6^ cells/20 min).	EQRLGHQWAVGHLM	[98]
Bombesin-related peptide	*Bombina variegate* (Yellow-bellied toad)	–	EDSFGNQWARGHFM	[98]
Brevinin-1CBb	*Lithobates septentrionalis* (Mink frog)	Insulin-releasing activity (in vitro): BRIN-BD11 clonal β-cells (1.0–3.0 ng/10^6^ cells/20 min). LDH-releasing activity (in vitro): BRIN-BD11 clonal β-cells (101.5–144.3% of control).	FLPFIARLAAKVFPSIICSVTKKC	[98,101]
Brevinin-1Pa	*Rana pipiens* (Northern leopard frog)	Insulin-releasing activity (in vitro): BRIN-BD11 clonal β-cells.	FLPIIAGVAAKVFPKIFCAISKKC	[98]
Brevinin-1E	*Rana esculenta* (Green frog)	Insulin-releasing activity (in vitro): BRIN-BD11 clonal β-cells.	FLPAIFRMAAKVVPTIICSITKKC	[98]
Brevinin-2GUb	*Hylarana guentheri* (Günther’s frog)	Insulin-releasing activity (in vitro): BRIN-BD11 clonal β-cells (0.7–1.8 ng/10^6^ cells/20 min).	GVIIDTLKGAAKTVAAELLRKAHCKLTNSC	[98,102]
Brevinin-2EC	*Pelophylax esculentus* (European black-spotted frog)	Insulin-releasing activity (in vitro): BRIN-BD11 clonal β-cells.	GILLDKLKNFAKTAGKGVLQSLLNTASCKLSGQC	[98]
Brevinin-2-related peptide (B2RP)	*Lithobates septentrionalis* (Mink frog)	Insulin-releasing activity (in vitro): BRIN-BD11 clonal β-cells.	GIWDTIKSMGKVFAGKILQNL	[98]
	*Lithobates septentrionalis* (Mink frog)	Insulin-releasing activity (in vitro): BRIN-BD11 clonal β-cells.	GMASKAGSVLGKVAKVALKAAL	
Caerulein-B1	*Xenopus borealis* (Marsabit clawed frog)	Insulin-releasing activity (in vitro): BRIN-BD11 clonal β-cells (1.4 ± 0.1 ng/10^6^ cells/20 min, stimulated rate 1.8 ± 0.0 ng/10^6^ cells/20 min).	EQDY(SO3)GTGWMDF	[98,103]
CPF-AM1	*Xenopus amieti* (Volcano clawed frog)	GLP-1-releasing activity by GLUTag cells (in vitro): 3.2-fold greater than basal rate at 3 μM concentration).	GLGSVL GKALKIGANLL-NH_2_	[14]
Dermaseptin B4	*Phyllomedusa trinitatis* (Trinidadian monkey frog)	Insulin-releasing activity (in vitro): BRIN-BD11 clonal β-cells.	ALWKDILKNVGKAAGKAVLNTVTDMVNQ	[98]
Dermaseptin-LI1	*Agalychnis litodryas* (Red-eyed treefrog)	Insulin-releasing activity (in vitro): BRIN-BD11 clonal β-cells.	AVWKDFLKNIGKAAGKAVLNSVTDMVNE	[98]
Esculentin-1	*Rana esculenta* (Green frog)	Insulin-releasing activity (in vitro): BRIN-BD11 clonal β-cells (3.0 ± 0.3 ng/10^6^ cells/20 min).	GIFSKLGRKKIKNLLISGLKNVGKEVGMDVVRTGIDIAGCKIKGEC	[98,104]
Esculentin-1b	*Rana esculenta* (Green frog)	Insulin-releasing activity (in vitro): BRIN-BD11 clonal β-cells (2.9 ± 0.2 ng/10^6^ cells/20 min).	GIFSKLAGKKLKNLLISGLKNVGKEVGMDVVRTGIDIAGCKIKGEC	[98,104]
Esculentin-2Cha	*Lithobates chiricahuensis* (Chiricahua leopard frog)	Insulin-releasing activity (in vitro): BRIN-BD11 clonal β-cells (TC: 0.3 nM). LDH-releasing activity (in vitro): BRIN-BD11 clonal β-cells (102.7 ± 1.1% of control). Insulin-releasing activity in high fat fed mice (in vivo): Plasma insulin (AUC: 165.0 ± 16. 9 ng/mL/min).	GFSSIFRGVAKFASKGLGKDLAKLGVDLVACKISKQC	[105]
GM-14	*Bombina variegate* (Yellow-bellied toad)	Insulin-releasing activity (in vitro): BRIN-BD11 clonal β-cells.	GKPFYPPPIYPEDM	[98]
GPPGPA	*Andrias davidianus* (Chinese giant salamander)	α-glucosidase inhibitory activity (in vitro): IC_50_ (0.3 ± 0.1 mg/mL).	Unknown.	[99]
Hymenochirin-1B	*Hymenochirus boettgeri* (Zaire dwarf clawed frog)	Insulin-releasing activity (in vitro): BRIN-BD11 clonal β-cells (TC: 1.0 nM, Insulin release at 1 μM: 13.4 ± 0.1% of total insulin content, Basal release of insulin at 1 nM: 304.4 ± 19.4%).	IKLSPETKDNLKKVLKGAIKGAIAVAKMV	[98,106]
IN-21	*Bombina variegate* (Yellow-bellied toad)	Insulin-releasing activity (in vitro): BRIN-BD11 clonal β-cells.	IYNAICPCKHCNKCKPGLLAN	[98]
Magainin-AM1	*Xenopus amieti* (Volcano clawed frog)	GLP-1-releasing activity (in vitro): GLUTag cells (1.1–2.0 pg/10^6^ cells/h).	GIKEFAHSLGKFG KAFVGGILNQ	[14]
Magainin-AM2	*Xenopus amieti* (Volcano clawed frog)	GLP-1-releasing activity (in vitro): GLUTag cells (1.1–2.7 pg/10^6^ cells/h).	GVSKILHSAGKFGKAFLGEIMKS	[14]
Ocellatin-L2	*Leptodactylus laticeps* (Santa Fe white-lipped frog)	Insulin-releasing activity (in vitro): BRIN-BD11 clonal β-cells (181% of basal rate at 3 μM).	GVVDILKGAAKDLAGHLATKVMDKL	[98,107]
Palustrin-1c	*Lithobates palustris* (Pickerel frog)	–	ALSILRGLEKLAKMGIALTNCKATKKC	[98]
Palustrin-2CBa	*Rana catesbeiana* (American bullfrog)	Insulin-releasing activity (in vitro): BRIN-BD11 clonal β-cells (1.1–2.4 ng/10^6^ cells/20 min). LDH-releasing activity (in vitro): BRIN-BD11 clonal β-cells (100.3–104.4% of control).	GFLDIIKDTGKEFAVKILNNLKCKLAGGCPP	[98,101]
PGLa-AM1	*Xenopus amieti* (Volcano clawed frog)	GLP-1-releasing activity (in vitro): GLUTag cells (1.1–2.6 pg/10^6^ cells/h).	GMASKAGSVLGKVAKVALKAAL-NH_2_	[14]
Phylloseptin-L2	*Hylomantis lemur* (Lemur tree frog)	Insulin-releasing activity (in vitro): BRIN-BD11 clonal β-cells (2.2 ± 0.2 ng/10^6^ cells/20 min). Insulin-releasing activity in mice (in vivo): Plasma insulin (AUC: 59 ng/mL/min).	FLSLIPHVISALSSL	[98,108]
Plasticin-L1	*Leptodactylus laticeps* (Santa Fe white-lipped frog)	Insulin-releasing activity (in vitro): BRIN-BD11 clonal β-cells (1.0–1.7 ng/10^6^ cells/20 min). LDH-releasing activity (in vitro): BRIN-BD11 clonal β-cells (101.1–113.7% of basal release).	GLVNGLLSSVLGGGQGGGGLLGGIL	[98,107]
Pseudhymenochirin-1Pb	*Pseudhymenochirus merlini* (Merlin’s dwarf gray frog)	Insulin-releasing activity (in vitro): BRIN-BD11 clonal β-cells (1.0–5.0 ng/10^6^ cells/20 min). LDH-releasing activity (in vitro): BRIN-BD11 clonal β-cells (460% of control at 3 μM).	GIFPIFAKLLGKVIKVASSLISKGRTE	[98,109]
Pseudhymenochirin-2Pa	*Pseudhymenochirus merlini* (Merlin’s dwarf gray frog)	Insulin-releasing activity (in vitro): BRIN-BD11 clonal β-cells (1.0–7.0 ng/10^6^ cells/20 min). LDH-releasing activity (in vitro): BRIN-BD11 clonal β-cells (<1500% of control at 3 μM).	IKIPSFFRNILKKVGKEAVSLIAGALKQS	[98,109]
Pseudin-2	*Pseudis paradoxa* (Shrinking frog)	Insulin-releasing activity (in vitro): BRIN-BD11 clonal β-cells (1.0–2.0 ng/10^6^ cells/20 min). LDH-releasing activity (in vitro): BRIN-BD11 clonal β-cells (100–150% of control).	GLNALKKVFQGIHEAIKLINNHVQ	[98,110]
Ranatuerin-1CBa	*Rana catesbeiana* (American bullfrog)	–	SMLSVLKNLGKVGLGFVACKVNKQC	[98]
Ranatuerin-2CBc	*Rana catesbeiana* (American bullfrog)	Insulin-releasing activity (in vitro): BRIN-BD11 clonal β-cells (1.0–2.8 ng/10^6^ cells × 20 min). LDH-releasing activity (in vitro): BRIN-BD11 clonal β-cells (98.8–103.7% of control).	GFLDIIKNLGKTFAGHMLDKIKCTIGTCPPSP	[98,101]
Ranatuerin-2CBd	*Rana catesbeiana* (American bullfrog)	Insulin-releasing activity (in vitro): BRIN-BD11 clonal β-cells (1.2–2.5 ng/10^6^ cells × 20 min). LDH-releasing activity (in vitro): BRIN-BD11 clonal β-cells (99.2–103.8% of control).	GFLDIIKNLGKTFAGHMLDKIRCTIGTCPPSP	[98,101]
RK-13	*Agalychnis calcarifer* (Costa Rican flying tree frog)	Insulin-releasing activity (in vitro): BRIN-BD11 clonal β-cells (0.8–1.5 ng/10^6^ cells × 20 min).	RRKPLFPLIPRPK	[98,111]
Temporin-CBa	*Rana catesbeiana* (American bullfrog)	Insulin-releasing activity (in vitro): BRIN-BD11 clonal β-cells (0.9–2.1 ng/10^6^ cells × 20 min). LDH-releasing activity (in vitro): BRIN-BD11 clonal β-cells (101.2–104.8 % of control).	FLPIASLLGKYL	[98,101]
Temporin-CBf	*Rana catesbeiana* (American bullfrog)	Insulin-releasing activity (in vitro): BRIN-BD11 clonal β-cells (1.0–2.4 ng/10^6^ cells × 20 min). LDH-releasing activity (in vitro): BRIN-BD11 clonal β-cells (101.2–103.9% of control).	FLPIASMLGKYL	[98,101]
Temporin-DRa	*Rana draytonii* (California red-legged frog)	Insulin-releasing activity (in vitro): BRIN-BD11 clonal β-cells (0.6–0.9 ng/10^6^ cells × 20 min). LDH-releasing activity (in vitro): BRIN-BD11 clonal β-cells (3.5–6.4% of total cell content).	HFLGTLVNLAKKIL	[98,112]
Temporin-DRb	*Rana draytonii* (California red-legged frog)	Insulin-releasing activity (in vitro): BRIN-BD11 clonal β-cells (0.6–1.5 ng/10^6^ cells × 20 min). LDH-releasing activity (in vitro): BRIN-BD11 clonal β-cells (3.7–5.7% of total cell content).	NFLGTLVNLAKKIL	[98,112]
Temporin-Oe	*Rana ornativentris* (Montane brown frog)	Insulin-releasing activity (in vitro): BRIN-BD11 clonal β-cells (0.7–2.3 ng/10^6^ cells × 20 min). LDH-releasing activity (in vitro): BRIN-BD11 clonal β-cells (6.2–9.2% of total cell content).	ILPLLGNLLNGLL	[98,112]
Temporin-TGb	*Rana tagoi* (Tago’s brown frog)	Insulin-releasing activity (in vitro): BRIN-BD11 clonal β-cells (0.6–1.8 ng/10^6^ cells × 20 min). LDH-releasing activity (in vitro): BRIN-BD11 clonal β-cells (4.0–5.6% of total cell content).	AVDLAKIANKVLSSLF	[98,112]
Temporin-Va	*Lithobates virgatipes* (Carpenter frog)	Insulin-releasing activity (in vitro): BRIN-BD11 clonal β-cells (0.8–1.5 ng/(10^6^ cells × 20 min). LDH-releasing activity (in vitro): BRIN-BD11 clonal β-cells (7.4–13.6% of total cell content).	FLSSIGKLIGNLL	[98,112]
Temporin-Vb	*Lithobates virgatipes* (Carpenter frog)	Insulin-releasing activity (in vitro): BRIN-BD11 clonal β-cells (0.8–1.7 ng/10^6^ cells × 20 min). LDH-releasing activity (in vitro): BRIN-BD11 clonal β-cells (7.4–9.8% of total cell content).	FLSIIAKVLGSLF	[98,112]
Temporin-Vc	*Lithobates virgatipes* (Carpenter frog)	Insulin-releasing activity (in vitro): BRIN-BD11 clonal β-cells (0.8–2.6 ng/10^6^ cells × 20 min). LDH-releasing activity (in vitro): BRIN-BD11 clonal β-cells (7.4–10.9% of total cell content).	FLPLVTMLLGKLF	[98,112]
Tigerinin-1R	*Rana tigerina* (Asian bullfrog)	GLUTag cell culture (in vitro), glucose homeostasis and beta cell function in mice with diet-induced obesity-diabetes (in vivo).	RVCSAIPLPICH	[14,24]
Xenopsin	*Xenopus amieti* (Volcano clawed frog)	Insulin-releasing activity (in vitro): BRIN-BD11 clonal β-cells (1.5–2.1 ng/10^6^ cells/20 min).	EGKRPWIL	[98,103]
Xenopsin-AM2	*Xenopus amieti* (Volcano clawed frog)	Insulin-releasing activity (in vitro): BRIN-BD11 clonal β-cells (1.0–2.0 ng/10^6^ cells/20 min).	EGRRPWIL	[98,103]

AUC: area under curve; TC: threshold concentration (minimum concentration of peptide producing a significant (*p* < 0.05) increase in the rate of insulin release); Basal release: the rate of insulin release in the absence of peptide and is set at 100%; IC_50_: half maximal inhibitory concentration; GLP-1: glucagon-like peptide-1; ns.: not significant; –: not specified.

## Data Availability

Not applicable.

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
