# Peer review of "Amphibian Skin and Skin Secretion: An Exotic Source of Bioactive Peptides and Its Application"

_foods, 2023, doi:10.3390/foods12061282_

Round 1
Reviewer 1 Report
In this paper the authors have reviewed in the first part the literature concerning the biological effects of bioactive peptides (BP) deriving from the skin and amphibian secretions. In the last part, the authors indicated the possible limitations (ethical/social...) of the exploitation of BP from these food by-products and the possible applications.
Although the topic is of interest, this review has some limitations that need to be improved before being accepted for publication.
Major revisions:
Paragraphs 1 and 2 are outside the scope of this review which focuses on the biological effects of bioactive peptides. I suggest deleting these unnecessary paragraphs and modifying the text of the manuscript accordingly.
Too much unnecessary pleonastic information that has already been extensively described. I refer for example to the description of ROS, the mechanisms of action of bioactive peptides and more. This part is unnecessary and already extensively described in other reviews.
The authors should describe much more intensively and extensively the studies of the biological effects of BP. For example, in the part relating to the antioxidant action, in the text only refs 15, 48 and 54 are described while other studies are reported in the table. In addition, the tables must provide more information such as the type of assay used, whether it was conducted in vitro or in cell cultures, the degree of antioxidant activity, the part from which the bioactive peptide was taken and more. The increase in information on the effects of BP should also be extended to the anti-microbial and anti-cancer effect...
In the exploitation part, the authors focused more on collagen and not on bioactive peptides.
One of the main limitations in the study of the biological effects of bioactive peptides present in foods or their by-products is the determination of these effects without considering the modifications that may occur to the peptide in terms of hydrolytic cuts during digestion. In two recent publications (10.3390/ijms232012555; 10.3390/molecules27030664), it has been seen how peptide sequences present in food are degraded during in vitro digestion and new sequences are formed. Therefore, the sequences initially present in food may probably not be present at the end of digestion. The authors should consider and describe this event that could limit the various studies on the biological effects of peptides if they have not first undergone a digestion process.
Reviewer 2 Report
The manuscript includes an interesting review focused on the employment of amphibian skin for obtaining active peptides. The study is well presented, justified and provides useful information. I provided some minor aspects are performed or clarified.
Abstract
The authors ought to mention the possible valuable functions of BP compounds.
Main parts of the manuscript
Lines 200-249: Too much information related to general aspects of lipid oxidation development. Such information could be shortened. It is at line 249 that really starts the item of the current manuscript.
Similarly, line 273-300 is also general information that ought to be notably shortened. It is in line 306 that the main item of the manuscript starts.
Similar comments could be done when explaining general aspects for anticancer and antidiabetic activities.
Line 494: executed.
Conclusions
Rather than Conclusions, the authors ought to include a section related to “further research, on-coming research or future trends”. After all, the title includes “future application” and this has really been addressed very scarcely.
Round 2
Reviewer 1 Report
The revised version of the paper has been improved by the Authors but I have doubts how the authors expressed the antioxidant activity in table 2. In fact, ABTS, DPPH and FRAP in table 2 are expressed as % when normally the value of this assay is indicated as mM of Trolox equivalent. What is referred to the %? Why did they not express as Trolox equivalent? To corroborate the unexpected value, the Authors indicated in the text the IC50 in micromolar concentration
Even in table 2 for ref 74 and 31, the authors indicated the effect on biological antioxidant and NO production without giving information on the type of model (human, animal, cell culture) and the concentration of peptides supplemented. The authors should include this information
In addition, the Authors should explain why the first three peptides have no assays and correct the acronym FRP to FRAP in the table
